# Nutritional Considerations for Peripheral Arterial Disease: A Narrative Review

**DOI:** 10.3390/nu11061219

**Published:** 2019-05-29

**Authors:** Christopher L. Delaney, Matilda K. Smale, Michelle D. Miller

**Affiliations:** 1Department of Vascular Surgery, Flinders Medical Centre and Flinders University, Bedford Park 5042, South Australia, Australia; 2Department of Nutrition and Dietetics, College of Nursing and Health Sciences, Flinders University, Bedford Park 5042, South Australia, Australia; smal0075@flinders.edu.au (M.K.S.); michelle.miller@flinders.edu.au (M.D.M.)

**Keywords:** atherosclerosis, peripheral arterial disease, malnutrition

## Abstract

Those with peripheral arterial disease (PAD) require important considerations with respect to food and nutrition, owing to advanced age, poor diet behaviours and immobility associated with the disease process and co-morbid state. These considerations, coupled with the economic effectiveness of medical nutrition therapy, mandate that dietetic care plays a vital role in the management of PAD. Despite this, optimising dietetic care in PAD remains poorly understood. This narrative review considers the role of medical nutrition therapy in every stage of the PAD process, ranging from the onset and initiation of disease to well established and advanced disease. In each case, the potential benefits of traditional and novel medical nutrition therapy are discussed.

## 1. Introduction

Those with peripheral arterial disease (PAD) require important considerations with respect to food and nutrition, and sub-optimal nutritional status in these patients is well documented [1]. The disease process and associated co-morbidities can lead to poor quality diet and low levels of physical activity, sometimes complete immobility, with resultant energy, protein and micronutrient deficiencies. In some cases, sarcopenia or sarcopenic obesity may occur [2], where deficiencies are masked by overweight and obesity [3]. Literature, including our own recent work, has demonstrated that up to 78% of vascular surgical patients could be classified as malnourished [1].

Sub-optimal nutritional status of patients with PAD is worsened by the lack of awareness displayed by physicians, dietitians and patients themselves regarding the need for medical nutrition therapy to complement all stages of management of the vascular disease process.

In the primary care setting, a study of 633 General Practitioners (GP) in Europe demonstrated that more than 50% of GP’s do not provide any dietary advice to patients. A lack of education in dietary issues and time to address these issues were cited as major barriers to the provision of such advice, as was the perception that many patients are unwilling to accept advice [4].

From a surgical perspective, peri-operative nutrition has recently been described as a “surgical orphan”, with work demonstrating that only 20% of surgeons are prepared to initiate routine nutritional screening and support according to evidence based guidelines [5]. Financial and logistic restrictions were considered to be the limitations to appropriate nutritional assessment, despite the fact that up to 80% of these surgeons made the link between malnutrition and poor peri-operative outcomes.

Perhaps the root of the problem is captured by work investigating the state of nutritional education at medical schools in the United States. This concluded that exposure to nutritional education during training is so poor that we cannot expect physicians to effectively identify and treat nutrition related diseases [6].

While the above should be considered a failure of current medical education strategies, what is more concerning is the variation in practice and associated clinical impact that is displayed by fully-trained dietitians involved in the treatment of patients with PAD. A questionnaire administered to Australian dietitians revealed significant variation in energy and protein estimates both, within and across stages of PAD. Analysis suggested that the variability of such estimates would be enough to have a clinically meaningful impact on patient outcomes. The limited evidence available to inform dietitians on best practice for PAD patients was proposed to be the likely explanation for the observed variability in practice [7].

Just as concerning are the findings of a large-scale European study of nearly 8000 patients in the primary care setting, designed to assess their attitudes towards lifestyle, nutrition and physical activity. The study concluded that a large proportion of patients with unhealthy lifestyles do not perceive the need to make change [8]. Compounding this issue is the fact that patients with PAD have a poor understanding of their risk factors and the need for secondary prevention strategies and they rely heavily on the provision of knowledge and education from health professionals, which, as highlighted earlier, is variable, if it is addressed at all [9].

The following narrative review aims to provide clinicians with an overview of the role of nutritional support in every stage of PAD, ranging from the onset and initiation of disease to the spectrum of established disease. In each case, the potential benefits of traditional and novel medical nutrition therapies will be discussed.

## 2. Nutrition in the Prevention of Endothelial Dysfunction and Atherosclerosis

Peripheral arterial disease is a local manifestation of the diverse pathophysiological processes associated with the systemic disease state atherosclerosis, however, both are preceded by the pathophysiological state known as endothelial dysfunction.

The vascular endothelium is a single, monolayer of cells acting as a barrier between blood and a pro-thrombotic vessel wall. It is a dynamic organ that plays a key role in vascular homeostasis and systemic well-being. Endothelial dysfunction is a state of impairment of these regulatory functions, characterised by an impaired endothelium dependent vasodilatory response to changes in flow or stimuli and a systemic pro-inflammatory and pro-oxidative state [10]. Endothelial dysfunction may be precipitated by genetic factors, such as hypercoagulability and blood group, and it has been proposed as the mechanism linking cardiovascular risk factors to established disease. Its onset has been deemed a sentinel event in the progression to atherosclerosis [11]. Modulation of endothelial function is, therefore, the likely mechanism through which medical nutrition therapy, either those targeting diet responsive risk factors or the systemic pro-inflammatory and pro-oxidative state of individuals with atherosclerosis, have been shown to delay the onset and progression of atherosclerosis [11,12].

Diet responsive cardiovascular risk factors include obesity, hypertension, dyslipidemia and diabetes mellitus.

Caloric restriction resulting in a negative energy balance has been the traditional approach to combat obesity. While this may be achieved in a variety of ways, there is evidence to suggest this may be best achieved in the form of an energy-restricted, high protein, low fat diet. This is associated with the greatest improvement in cardiovascular risk profile [13].

The DASH (dietary approaches to stop hypertension) diet is a dietary pattern rich in fruit, vegetables, nuts and fibre. Considered by many a major advance in nutritional science, its effect may be augmented by low sodium intake and it has been shown to consistently lower blood pressure throughout the literature [14].

With respect to hyperglycaemia, a position statement from the American Diabetes Association [15] recommends that medical nutrition therapy is important in both prevention and management. Specifically, monitoring carbohydrate intake remains a key strategy in achieving glycaemic control, while saturated fat should be limited to <7% of total caloric intake. The omega-3 polyunsaturated fatty acids derived from at least two fish meals per week are also recommended as beneficial due to their capacity to modulate insulin resistance [16].

Although a recent systematic review suggested that omega-3 fatty acids are not useful in the management of established cardiovascular disease, they can be an effective method of risk factor modification [17]. Specifically, omega-3 fatty acids have been shown to have a weak anti-hypertensive effect as well as an ability to reduce triglyceride levels and improve lipid profile [12,17,18]. This may, in part, explain why populations with a high dietary intake of omega-3 fatty acids have low reported levels of cardiovascular disease [19], however, the lipid lowering capacity of dietary intervention is not limited to omega-3 fatty acids. As demonstrated in a meta-analysis, soluble fibres are capable of lowering both total and LDL cholesterol levels, while polyphenols, such as resveratrol, have a lipid lowering effect in vitro but at this stage, such an effect has not translated into a clinical benefit on lipid profile [20].

Interestingly, red wine is a good source of the polyphenol resveratrol and moderate alcohol intake may be protective against cardiovascular disease [21]. The adverse effects of alcohol consumption mean that alcohol intake is not currently a recommended dietary strategy, however, the strong anti-oxidant properties of polyphenols and their ability to modulate the systemic pro-inflammatory and pro-oxidative state associated with endothelial dysfunction and atherosclerosis [22] means that polyphenols may contribute to the observed athero-protective effect of alcohol [22]. The capacity of medical nutrition therapy to temper this systemic state may, therefore, be just as important as dietary modification of risk factors.

In addition to polyphenols, nutrients such as omega-3 fatty acids, L-arginine, folic acid, cacao and green-tea have all independently been shown to have a positive effect on endothelial health [23]. In each case, the mechanism of effect is likely to be slightly different. It may be either an anti-oxidant effect to combat the oxidative stress, often triggered by cardiovascular risk factors, that is believed to be a causative factor in the onset of endothelial dysfunction, or a direct metabolic effect leading to enhancement of endothelial nitric oxide secretion. Nitric oxide itself is an important molecule released by the endothelium, which acts as a potent vasodilator with strong anti-oxidant and anti-inflammatory properties [24].

Vitamin D deficiency has also been linked to endothelial dysfunction and a small uncontrolled intervention study revealed an improvement in markers of endothelial health following vitamin D supplementation in a cohort of patients with ischaemic heart disease [25]. Additionally, data from the National Health and Nutrition Examination Survey (NHANES) in the United States also showed vitamin D deficiency to be associated with a greater prevalence of PAD, however it is unclear whether such a relationship is causative [26].

Not only are individual nutrients capable of modifying cardiovascular risk factors and augmenting endothelial function, they also have the capacity to exert beneficial athero-protective effects at various stages of the development and progression of atherosclerotic plaque. Nutrients such as hydroxytyrosol found in olive oil and flavonols demonstrate anti-oxidant capacity, which can prevent the oxidation of lipoprotein particles and subsequent fatty streak formation [27]. The amino-acid like compound taurine exerts a similar effect through its ability to inhibit expression of the oxidised LDL-receptor [28]. Taurine, like vitamin E and omega-3 fatty acids, has also been shown to reduce expression of endothelial cellular adhesion molecules with a resultant decrease of leukocyte migration into the intima and impairment of foam cell formation [28]. Vitamin E may also contribute to plaque stability by stimulating the synthesis of extracellular matrix and fibrosis [28], however, like many of these nutrients, the evidence is not currently strong enough to recommend routine supplementation.

An often overlooked vitamin in the pathogenesis of atherosclerosis is vitamin K. Arterial calcification occurs as part of the atherosclerotic process but may be inhibited by a vitamin K dependent protein known as matrix GIa protein (MGP) [29]. Antagonism of the vitamin K cycle results in high levels of uncarboxylated MGP and more severe arterial calcification. It is, therefore, plausible that vitamin K supplementation may slow the progression of calcification and atherosclerosis and work is being undertaken to investigate this. Phylloquinone (vitamin K1) is the primary dietary source of vitamin K, however, menaquinone (vitamin K2) may have a greater association with vascular calcification, and is therefore the preferred supplement for current trials [30].

More recently, the focus has shifted from individual nutrients to dietary patterns that may be capable of combating the systemic state associated with endothelial dysfunction, and subsequently, atherosclerosis. Broadly speaking, dietary patterns rich in fruit, vegetables, fish and nuts have a beneficial impact on endothelial function and cardiovascular disease progression and encompass the commonly referred to “Mediterranean diet” [12] and the DASH diet [14], as well as the high levels of omega-3 fatty acid consumption associated with the “Seafood diet” [31]. In contrast, the intake of processed meats, fried food and refined grain products contribute to the so called “Western diet”, which is pro-inflammatory and pro-atherogenic [32].

Advances in nutritional science have fostered the evolution of a novel dietary strategy, referred to as “immuno-nutrition” [33]. Referring to the potential to modulate the immune system by interventions with specific nutrients, the concept of immuno-nutrition allows for a synergistic approach to immune-modulation through combined supplementation of anti-oxidant and anti-inflammatory vitamins and minerals [34], with similar properties of amino-acids, such as arginine [35] and glutamine, as well as omega-3 polyunsaturated fatty acids [36]. The subsequent enhancement of immune function has been shown to suppress acute post-operative inflammatory responses and protect against post-operative complications in patients undergoing major abdominal surgery [37]. Additionally, a meta-analysis reported a significant reduction in infection rate and hospital length of stay among general surgical patients with peri-operative immuno-nutrition supplementation [34]. Despite these promising findings, the immunosuppression associated with malnutrition in many patients with atherosclerotic disease and the fact that both innate and adaptive immune responses have been implicated in the chronic inflammatory state of atherosclerosis, no work has been undertaken to investigate the capacity of immuno-nutrition to slow or even halt the progression of atherosclerosis and associated cardiovascular event profile. This represents an opportunity for future clinical and translational research which may be complementary to current work that is targeting the immune-mediated components of atherosclerosis in an attempt to favourably alter the natural history of the disease process in the form of a vaccination.

In the meantime, to complement lifestyle modifications, such as tobacco avoidance and regular physical activity, the use of medical nutrition therapy to treat cardiovascular risk factors and the systemic state associated with atherosclerosis should be initiated in the primary care setting (see Table 1) and continued at a specialist level for those patients with clinical manifestations of the disease process. Consideration should also be given to use of such supplementation as a primary preventative strategy to augment endothelial function and potentially prevent progression to atherosclerosis in younger individuals whose genotype may place them at high risk for future disease.

## 3. Nutrition in the Management of Peripheral Arterial Disease

Peripheral arterial disease is an occlusion or stenosis of an artery, usually one belonging to the leg. It occurs across a spectrum of disease ranging from asymptomatic to critical limb ischaemia (characterised by ischaemic rest pain and tissue loss) [38]. The nutritional requirements and medical nutrition therapies for these patients vary depending on the severity of disease.

### 3.1. Asymptomatic PAD

Up to 25% of patients with PAD are asymptomatic, many of whom remain unaware of the diagnosis until it is incidentally detected during the course of investigating another pathology [39]. Despite this, the risk of associated cardiovascular morbidity and mortality is equivalent to those with symptoms. That is, 20% of these patients will suffer from a non-fatal cardiovascular (CV) event within 5 years, while 10–15% will be dead within 5 years, the majority of these from a CV related cause [40]. For those who are aware of the diagnosis, treatment strategies, therefore, involve halting progression of disease and atherosclerotic plaque stabilisation to avoid CV events. This is achieved through lifestyle intervention and pharmacotherapies to modify risk factors and limit the systemic pro-inflammatory and pro-oxidative state of atherosclerosis [41].

Implementation of medical nutrition therapies described above to prevent progression and onset of atherosclerosis are paramount, however, given these patients have established disease, prescription of what is known as “best medical therapy”, typically comprising of an antiplatelet agent, a statin and anti-hypertensive drugs, is also a vital component of the treatment paradigm. Consideration must, therefore, be given to the concept of drug–nutrient interactions. A review article which critically appraised the evidence exploring the association between medications commonly prescribed to PAD patients and nutritional status highlighted a risk of Coenzyme Q10 deficiency with lipid lowering medications, zinc depletion with anti-hypertensive medications and vitamin B12 deficiency with oral hypoglycaemic agents. It was concluded that patients with PAD, who are long term consumers of such medications, should be routinely monitored for these nutrient deficiencies to facilitate early identification and initiation of treatment [42].

In addition to standard “best medical therapy”, anticoagulation in the form of warfarin is also commonly prescribed to these patients for the treatment of associated co-morbidities. In light of recent work considering the link between vascular calcification and vitamin K, it is possible that antagonism of vitamin K with warfarin therapy may contribute to more significant levels of calcification and disease progression. Therefore, alternative drugs, such as Apixiban or Rivoroxiban, should be considered if anticoagulation is necessary in patients with PAD. Care should also be taken with supplementation of vitamin E in these patients. Although not a drug–nutrient interaction, vitamin E supplementation may be antagonistic to vitamin K with the potential to impact the burden of calcification [43].

### 3.2. Intermittent Claudication

Intermittent claudication (IC) is the most frequent symptom of PAD and is defined as walking induced pain or cramping in one or both legs that is relieved by rest. In addition to achieving control of disease progression and CV risk reduction, treatment is also geared towards providing symptom relief by way of improvements in pain free walking performance. Supervised, treadmill-based exercise training is the current gold-standard of treatment to improve walking performance in this cohort of patients [44]. The influence of such training on the nutritional status of patients with IC has been poorly addressed in the literature. This is despite the potential consequences associated with increasing metabolic demands through high-intensity exercise without the concurrent administration of adequate nutrition in this already nutritionally vulnerable group [45,46]. In fact, evidence exists to suggest that combined nutrition and exercise interventions in the rehabilitation setting are preferable, with those receiving exercise alone demonstrating greater declines in nutritional status and physical health [47,48,49]. The timing and composition of nutritional support, in particular supplementary protein, is critical in achieving optimal outcomes for exercise interventions [50,51]. Nutrients with anti-oxidant and anti-inflammatory properties are also essential, not only to treat the underlying atherosclerotic systemic disease state, but also in light of recent work in patients with IC demonstrating a negative association between walking performance and markers of endothelial cell inflammation and a positive association with ambulatory ability and circulating anti-oxidant capacity [52].

To put this into context, patients with IC have demonstrated high intake of pro-atherogenic foods including saturated fat, sodium and cholesterol, and low intake of potentially anti-inflammatory and anti-oxidative foods, such as fibre, vitamins A, C, and E, and folate, compared with recommended daily dietary intake [3,53]. This may, in part, be attributable to the socio-economic status of these patients. A higher prevalence of PAD has been demonstrated in patients with lower socio-economic status [54] and social class is known to predict diet quality, with nutrient poor diets preferentially consumed by those with limited economic capacity [55]. Compounding this poor intake is the relative calf muscle atrophy in patients with IC compared with healthy controls, which may be worsened by an increased proteolytic effect associated with participation in a supervised exercise program [46]. The extent of muscle atrophy may not be limited just to the legs, with recent work suggesting that 25% of patients with IC have sarcopenia, with a greater associated impairment of walking performance than those with IC who are not sarcopenic [56].

The central role played by muscle in whole body protein metabolism is particularly important in response to physiological and pathological stress. Preservation of muscle mass is an important determinant of survival in chronic disease states, and the ability to perform activities of daily living, and therefore maintain quality of life, is also critically dependent on muscle mass [57]. Myokines, circulating hormones secreted by skeletal muscle, may even be able to influence characteristics of atherosclerotic plaque [58].

### 3.3. Critical Limb Ischaemia

Critical limb ischaemia (CLI) is the most advanced stage of PAD, with a prognosis worse than many cancer-related illnesses [59]. A high burden of atherosclerotic disease in association with significantly raised levels of inflammatory cytokines [60] and reduced total anti-oxidant capacity [61], when compared to patients with IC, leads to a 25% risk of mortality and 30% risk of major limb amputation within 12 months of diagnosis [40].

Contributing to the poor prognosis in this patient cohort is the high prevalence of malnutrition, recently reported to be as much as 78% [1]. Not surprisingly, those at the severe end of the spectrum of malnutrition are more likely to die within 30 days of diagnosis [62]. Advanced age and poor oral intake, immobility, co-morbidities, atherosclerotic disease burden, and in many cases, prolonged hospital admission with or without associated procedural intervention are all factors associated with a diagnosis of malnutrition. Specific, disease-related features may also be responsible and are somewhat unique to patients with PAD and CLI.

The term “Resting Energy Expenditure” (REE) represents the amount of calories utilised by an individual in an inactive state over a 24 h period. Metabolism of skeletal muscle is the major contributor to daily REE, and owing to the progressive disuse calf muscle atrophy and muscle denervation across the spectrum of PAD, it is not surprising that patients with CLI have a lower REE than those with IC, whose REE is lower than that of healthy controls [63,64]. When corrected for skeletal muscle mass, this observation remained and suggests that factors associated with peripheral blood flow also provide an important contribution to REE in these patients. In combination with the low level of physical activity undertaken by patients with CLI, these findings may represent an overall positive energy balance with subsequent risk of weight gain and obesity. In support of this statement are recent findings demonstrating a strong positive association between obesity and CLI [65]. Interestingly, in a concept known as “the obesity paradox”, a high body-mass index (BMI) may have a protective effect against mortality in patients with lower limb ulcers [66], however, this was a small study, in which 53% of patients had non-ischaemic ulcers (venous or neuropathic) and body composition was not assessed. Therefore, it is likely that preservation of skeletal muscle mass is also required for such a paradox to hold true. Instead, patients with CLI have muscle atrophy and any patient with obesity may be considered as having sarcopenic obesity. Effectively a combination of low muscle mass and high fat mass, sarcopenic obesity is an entity in which the consequences of sarcopenia and obesity are complementary and compound the negative pathophysiological effects of each other. In isolation, up to 44% of patients with PAD have sarcopenia [67], while 46% are obese [1]. The consequences of sarcopenia in patients with CLI include a greater risk of mortality, limb loss and CV events when compared to those without sarcopenia [68]. In addition, they experience decreased mobility, slow gait and poor physical endurance, which can exacerbate the clinical features associated with CLI. Excess adipose tissue may precipitate a pro-thrombotic state and platelet hypercoagulability, while also contributing to the chronic pro-inflammatory state of atherosclerosis and providing an impairment to timely wound healing [65].

The above discussion may not hold true for all patients with CLI, as the work to investigate REE in these patients only considered those with rest pain, excluding the presence of ulceration. In practice, the majority of patients with CLI present with ulceration. Isolated rest pain is relatively uncommon. There is currently no literature to describe the impact of ischaemic ulceration on REE, however, in a systematic review investigating energy balance in patients with pressure ulcers, patients were characterised by increased REE, which correlated with the surface area of the ulcer and reduced energy intake [69]. If this is proven to be correct in patients with CLI, a negative energy balance may exist resulting in progression of muscle atrophy and further deterioration of nutritional status. Preservation of skeletal muscle is essential to facilitate ulcer healing, as it is an important source of the protein required for collagen formation [70]. Protein deficiency contributes to poor wound healing and estimated protein requirements for patients attempting to heal wounds are typically too low [71]. An explanation for this under-estimation may be the daily protein loss that occurs from wound or ulcer exudate, which may be as much as 100 g per day [70]. Estimated energy requirements when using the Harris-Benedict equation for patients with pressure ulcers are also felt to be inadequate, with a correction factor of 10% recommended [69]. This may also be the case in patients with ischaemic ulceration and warrants further investigation [72]. Clinician education is, therefore, essential to ensure adequate dietary supplementation of protein and energy in the presence of tissue loss in CLI patients.

With respect to wound healing, other aspects of nutritional care are equally important. Micronutrients such as vitamin A, vitamin C and Zinc have anti-oxidant properties and are involved in wound healing and epithelial integrity, while selenium enhances wound healing mechanisms that are complementary to the role of zinc [73]. Vitamin D and vitamin E are also anti-oxidant and have immune-modulatory effects, which may indirectly augment the healing process. Higher rates of major limb amputation have been observed in CLI patients with vitamin D deficiency [74].

Despite the importance of these micronutrients, alarming rates of deficiency have been observed in the vascular surgery population in general, but specifically in those with PAD. In a recent study of 94 patients with PAD, 78% were vitamin C deficient, 58% were deplete in vitamin D and 50% displayed low levels of serum zinc [1]. The patient cohort also demonstrated widespread deficiencies in vitamin A, iron and vitamin B-12. Deficiencies in multiple micronutrients were commonly observed [1], but importantly, each individual demonstrated a unique pattern of micronutrient deficiency and nutritional requirements, meaning that supplementation must be tailored to meet the needs of each patient and cannot be generalised to suffice the patient cohort as a whole. In fact, unnecessary supplementation can lead to excess levels of these nutrients with associated adverse consequences [75].

## 4. Impact of the Perioperative State on Nutritional Status in PAD

In general terms, ischaemic ulceration or rest pain requires an intervention to revascularise the limb in order to achieve limb salvage and avoid amputation. In some cases, patients with IC also undergo revascularisation to treat lifestyle limiting symptoms and improve walking capacity.

The role of medical nutrition therapy in the peri-operative state has been well explored in the general surgical population and an inadequate nutritional state has been shown to increase length of stay, rate of hospital readmissions, healthcare costs, morbidity and mortality [34]. These findings are also applicable to vascular surgery, however, other features unique to the vascular surgical population also need consideration.

The high rates of sarcopenia and obesity in CLI patients, alone or in combination, which were previously referred to, are associated with poorer short and long-term post-operative outcomes, including mortality and limb salvage, when compared to those without sarcopenia or obesity [76].

Resting energy expenditure increases acutely after any surgical intervention, owing to the catabolic state precipitated by surgery and anaesthesia [34]. This can lead to progressive muscle atrophy and sarcopenia, compounding the nutritional vulnerability of this already at-risk population. Of equal relevance, however, is the impact of revascularisation on REE in the longer term. Although speculative, given that peripheral blood flow has been proposed to contribute to REE, probably due to improved perfusion and metabolic output of skeletal muscle, an increase in peripheral blood flow following revascularisation is likely to be associated with an increase in REE. The impact of this is to worsen what is already a potential negative energy balance, particularly in patients with ulcers, and potentially precipitate a further decline in nutritional state [77].

In cases of open surgical revascularisation, a long incision is generally required for purposes of conduit harvest and bypass. A catastrophic consequence of such a procedure is wound dehiscence, often in association with infection. Deep tissue infection with bypass graft involvement usually precedes graft explant and limb loss and the importance of timely wound healing is, therefore, paramount. From a nutritional perspective, as discussed above, appropriate supplementation of energy and protein as well as micronutrients is essential [34]. In the peri-operative phase, however, consideration must also be given to the fasting state, particularly the impact on glucose control in the high proportion of PAD patients with diabetes mellitus. Glucose is an important energy source for wound healing, however, hyperglycaemia can impair wound healing through enhancement of catabolism and impairment of necessary immune responses. Careful monitoring of blood glucose levels and intensive insulin therapy is, therefore, required alongside individualised medical nutrition therapy.

## 5. Long-Term Nutritional Requirements

It is important to remember that interventions for atherosclerosis and its associated clinical conditions are not curative. Patients will evolve through many stages of the disease process, in some cases showing signs of deterioration, and in other cases, after revascularisation for example, improve their clinical state. What is certain, however, is that once diagnosed with atherosclerotic disease, this is a chronic, lifelong diagnosis, as are the nutritional requirements that accompany it.

To highlight the importance of this statement is a concept that is often considered the “Achilles heel” of modern vascular surgery. Neo-intimal hyperplasia (NIH) is a fibro-proliferative response to mechanical injury of the wall, such as that induced by angioplasty, stenting or bypass surgery [78,79]. It commonly precipitates graft or stent failure and reinterventions to treat NIH, in order to maintain or restore patency, are both costly and resource heavy [80].

Given the immune-modulated proliferation and migration of vascular smooth muscle cells that is characteristic of NIH, current treatment strategies are directed towards achieving localised immunosuppression. Stents and angioplasty balloons are coated with drugs such as paclitaxel, whose anti-proliferative properties have been shown to improve patency rates of vascular interventions. Despite this, the cost of such treatment is relatively prohibitive and the safety profile has recently been brought into question [81].

Immuno-nutrition may represent a novel treatment approach to target NIH. Direct injection of vitamin D into stenotic lesions of patients with arterio-venous fistulae has been shown to inhibit the progression of NIH [82]. Similar findings have been reported in vitamin E treated rabbits [83], while augmentation of vitamin D receptors can reduce the incidence of NIH following coronary angioplasty in atherosclerotic swine [84]. Vitamins A and C [85] have also been proposed to reduce the rate of proliferation of endothelial and smooth muscle cells and biodegradable stents with built in vitamin A are being trialled to treat atherosclerotic lesions in patients with cardiovascular disease.

In addition to vitamins, endogenously generated omega-3 fatty acids can attenuate vascular inflammation and NIH through interaction with the free fatty acid receptor 4 in mice [86], as well as inhibit NIH in autologous vein grafts in dogs [87]. Combining the anti-oxidant and anti-inflammatory properties of the above-mentioned vitamins and nutrients, along with others such as zinc, arginine and glutamine, which are supplemented as part of immune-nutritional support, may therefore have a role to play in reducing the onset and progression of NIH with subsequent improvement in short and long-term patency rates of vascular interventions.

## 6. Summary and Recommendations

Patients with PAD are a nutritionally vulnerable group and the opportunity exists for targeted medical nutrition therapy to improve clinical outcomes across all stages of the disease process and potentially prevent disease onset in those at risk.

Improved education of physicians and allied health professionals involved in the care of patients with cardiovascular disease is required and this should be actively incorporated into both the under-graduate and post-graduate training of these individuals.

With such a significant proportion of patients with atherosclerosis demonstrating features of malnutrition, many of whom are not routinely identified, a formal nutritional assessment of all patients is recommended. It is, however, essential to tailor medical nutrition therapy to the unique needs of each individual patient. Micronutrients should be assessed for in serum samples and supplemented as required. Protein and energy requirements will vary depending on the presence or absence of ulcers, stage of disease, and whether or not a patient is in the peri-operative state or undergoing a concurrent exercise-based intervention. Traditional equations to calculate such requirements should be adapted accordingly (See Table 2).

Finally, medical nutrition therapy in general surgical patients has been demonstrated as an efficient investment, with every $1 of expenditure for hospitalised patients resulting in a saving of $52 [34]. On the basis of the above discussion, the potential reduction in health care expenditure for patients with PAD may be even greater and is worthy of significant research attention and investment.

## Figures and Tables

**Table 1 nutrients-11-01219-t001:** Diet and lifestyle recommendations for atherosclerosis prevention in the primary care setting.

**Lifestyle**	
	Avoid tobacco use
	Regular physical activity
	Minimum of 30 min moderate intensity physical activity at least 5 days/week
**Diet**	
Fats	
	Limit saturated fats to <7% of daily caloric intake
	Replace saturated fats with unsaturated fats (eg., omega-3 fatty acids)
	At least two fish meals per week
	Use olive oil for cooking
	High unsaturated fat content and the anti-oxidant hydroxytyrosol
	Consume low-fat dairy products
	Limit red meat to one meal per week
	Replace with beans or legumes
**Calories**	
	Limit sugar consumption (food and drink)
	Avoid excessive caloric intake from any source
**Sodium**	
	Limit intake to 1.7 g per day
**Fruit, Vegetables and Wholegrains**	
	Five serves of vegetables, two serves of fruit, four serves of wholegrain per day
	Ensures satisfactory daily fibre, vitamin and mineral intake
	Consume wholegrain in preference to white or refined grain products
**Anti-oxidants**	
	1–2 cups of green tea daily
	Fruit and vegetable intake as above

**Table 2 nutrients-11-01219-t002:** Recommendations for the nutritional assessment of patients with Peripheral Arterial Disease to determine their nutritional status and requirements.

**Assessment**	
	Given the high prevalence of malnutrition in PAD, all patients (inpatients or outpatients) with a diagnosis of PAD should undergo a formal dietitian assessment to determine nutritional status/requirements
	**Anthropometry**
	BMI, recent history of unintentional weight loss
	**Routine nutritional biochemistry**
	haemoglobin, iron studies, albumin and total protein, lipid profile, C-reactive protein and white cell count, glycated haemoglobin (HbA1c), electrolytes and creatinine
	**Micronutrient/Trace element screen**
	Vitamins A, C, D, E, Vitamin B12 and folate, Zinc
	**Clinical assessment**
	Stage of PAD (Asymptomatic, intermittent claudication or critical limb ischaemia)
	Peri-operative state or conservative management
	Potential drug:nutrient interactions
	**Dietary assessment**
	Does protein/energy intake match the estimated requirement?
	Adjust requirement if hypermetabolic state
	Variable adjustment depending on presence/size of ulceration; peri-operative state (magnitude of surgery performed); active infection
**Intervention**	
	Education/counselling/dietary recommendations to reduce risk of disease progression (see Table 1) and allow improved blood-glucose level control
	Tailor supplementation to meet the needs of individual patients
	Protein/Energy
	Micronutrients/Trace elements
**Monitoring**	
	Routine re-assessment of patients to identify a change in clinical state and associated nutritional requirements

PAD; Peripheral Arterial Disease, BMI; Body Mass Index.

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
