# Peer review of "Nutritional Considerations for Peripheral Arterial Disease: A Narrative Review"

_nutrients, 2019, doi:10.3390/nu11061219_

Round 1
Reviewer 1 Report
Overall comments: As it is a good narrative review with experts in this field, it would be informative to add two short text boxes based on data from this review; one general box with recommendation to prevent (primary prevention) atherosclerosis such as fish meals twice a week,….amount of fruits and vegetables…fibre intake, physical activity, no smoking…I am sure the authors can come up with good suggestions and firm advice for the genral practitioner, and one specific box with suggestion of formal assessment of nutritional status and requirements for the vascular surgery patient (for instance which blood analyses of nutrients are recommended without increasing costs to much and to be implemented as well as other analysis at admission such as hemoglobin, creatinine…and taking the most important other aspects into account such as the presence of a chronic foot ulcer). These text boxes would increase this papers readability.
Specific comments:
Line 207 ”than” not ”that”
Line 240. ”many patients had non-ischaemic ulcers”. Give the proportion. You mean venous leg ulcers? Please clarify or omit if this sentence not apply to the current review.
Author Response
Overall comments: As it is a good narrative review with experts in this field, it would be informative to add two short text boxes based on data from this review; one general box with recommendation to prevent (primary prevention) atherosclerosis such as fish meals twice a week,….amount of fruits and vegetables…fibre intake, physical activity, no smoking…I am sure the authors can come up with good suggestions and firm advice for the genral practitioner, and one specific box with suggestion of formal assessment of nutritional status and requirements for the vascular surgery patient (for instance which blood analyses of nutrients are recommended without increasing costs to much and to be implemented as well as other analysis at admission such as hemoglobin, creatinine…and taking the most important other aspects into account such as the presence of a chronic foot ulcer). These text boxes would increase this papers readability.
Thanks for your comments. We agree that such recommendations in text boxes would be useful and these have been included in the manuscript.
Specific comments:
Line 207 ”than” not ”that”: This change has been made
Line 240. ”many patients had non-ischaemic ulcers”. Give the proportion. You mean venous leg ulcers? Please clarify or omit if this sentence not apply to the current review.: The proportion with non-ischaemic ulcers was 53% and these were either venous or neuropathic. This has been updated in the manuscript
Reviewer 2 Report
The pathophysiology of atherosclerosis is complex, different mechanism involve intimal and media Changes and plaque growth/rupture .I Think the authors should better describe how these could be modified by nutrition and also vitamin supplements.
Genetics + infections are also important for atherosclerosis as is hypercoagulation /blood group (vWF levels).
Vitamin K is important for media atherosclerosis and has been linked to defect carboxylation (vitamin K dependant) of Matrix-Gla-protein in diabetes and dialysis patients and Morbus Keitel.
A supervitamin K2 - Menaquinone 7 is in the hot line for current big prospective studies.
Socioeconomics is very interlinked to nutrition - I would like the authors to better discuss this - this is a confounding factor in almost all prospective and retospective nutritional health effect studies.
Wine (red) and resveratrol - comments? Alcohol and atherosclerosis?
Medication + vaccine against atherosclerosis - current and future trends to prevent atherosclerosis or to make them go? Are these better than optimal food.
Omega3 is mentioned but there several recent studies negating any effect of this vitamin - comment?
Vitamin Ecan counteract Vitamin K2 -see above - comment?
Author Response
The pathophysiology of atherosclerosis is complex, different mechanism involve intimal and media Changes and plaque growth/rupture .I Think the authors should better describe how these could be modified by nutrition and also vitamin supplements.
More information on the role of specific nutrients/vitamins during different stages of atherosclerosis has been added to the manuscript as requested
Genetics + infections are also important for atherosclerosis as is hypercoagulation /blood group (vWF levels). This has now been mentioned in the manuscript
Vitamin K is important for media atherosclerosis and has been linked to defect carboxylation (vitamin K dependant) of Matrix-Gla-protein in diabetes and dialysis patients and Morbus Keitel.
A supervitamin K2 - Menaquinone 7 is in the hot line for current big prospective studies.
Thank you for your comments regarding vitamin K. The authors agree that this is an important area that was overlooked in the initial manuscript. The link between vitamin K and arterial calcification, as well as the role of vitamin K supplementation (specifically K2) has been appropriately added to the manuscript
Socioeconomics is very interlinked to nutrition - I would like the authors to better discuss this - this is a confounding factor in almost all prospective and retospective nutritional health effect studies.
Like vitamin K, the authors agree that socio-economic status is important. This has now been discussed in the manuscript
Wine (red) and resveratrol - comments? Alcohol and atherosclerosis? Red wine as a source of resveratrol and recommendations regarding alcohol and atherosclerosis have been added to the manuscript.
Medication + vaccine against atherosclerosis - current and future trends to prevent atherosclerosis or to make them go? Are these better than optimal food. The possibility that diet may be complementary to future immuno-modulatory vaccines targeting atherosclerosis has been mentioned in the manuscript
Omega3 is mentioned but there several recent studies negating any effect of this vitamin - comment? A systematic review has been referred to that discusses the fact that omega-3 is not useful in the management of established cardiovascular disease but has a beneficial role in risk factor modification.
Vitamin Ecan counteract Vitamin K2 -see above - comment? This has been discussed along with the drug:nutrient interaction of warfarin and vitamin K.
Round 2
Reviewer 2 Report
I an happy with the changes in the manuscript and the Tables and also with response to reviewrs comments.